# Structural basis for late maturation steps of the human mitoribosomal large subunit

Miriam Cipullo[1,2,5], Genís Valentín Gesé[3,5], Anas Khawaja[1,2], B. Martin Hällberg 🔾 [3,4✉] & Joanna Rorbach 🔾 [1,2✉]

Mitochondrial ribosomes (mitoribosomes) synthesize a critical set of proteins essential for oxidative phosphorylation. Therefore, mitoribosomal function is vital to the cellular energy supply. Mitoribosome biogenesis follows distinct molecular pathways that remain poorly understood. Here, we determine the cryo-EM structures of mitoribosomes isolated from human cell lines with either depleted or overexpressed mitoribosome assembly factor GTPBP5, allowing us to capture consecutive steps during mitoribosomal large subunit (mt-LSU) biogenesis. Our structures provide essential insights into the last steps of 16S rRNA folding, methylation and peptidyl transferase centre (PTC) completion, which require the coordinated action of nine assembly factors. We show that mammalian-specific MTERF4 contributes to the folding of 16S rRNA, allowing 16 S rRNA methylation by MRM2, while GTPBP5 and NSUN4 promote fine-tuning rRNA rearrangements leading to PTC formation. Moreover, our data reveal an unexpected involvement of the elongation factor mtEF-Tu in mt-LSU assembly, where mtEF-Tu interacts with GTPBP5, similar to its interaction with tRNA during translational elongation.

[1] Department of Medical Biochemistry and Biophysics, Division of Molecular Metabolism, Karolinska Institutet, Solna, Sweden. [2] Max Planck Institute Biology of Ageing—Karolinska Institutet Laboratory, Karolinska Institutet, Stockholm, Sweden. [3] Department of Cell and Molecular Biology, Karolinska Institutet, Solna, Sweden. [4] Centre for Structural Systems Biology (CSSB) and Karolinska Institutet VR-RÅC, Hamburg, Germany. [5]These authors contributed equally: Miriam Cipullo, Genís Valentín Gesé. ✉email: martin.hallberg@ki.se; joanna.rorbach@ki.se

Mitoribosomal function is essential for cellular energy supply and mitoribosomal defects are coupled to a large and diverse set of human diseases[1].

Mammalian mitoribosomes assemble in a multi-step process that includes the maturation of two ribosomal RNAs (rRNAs; 12S and 16S), a structural tRNA, and incorporation of 82 mitoribosomal proteins (MRPs)[2]. Several RNA-binding proteins, including helicases, GTP-binding proteins (GTPBPs), as well as RNA modifying enzymes have been shown to be essential for correct mitoribosome biogenesis[3]. Some of these factors are conserved in ribosome biosynthesis in all forms of life, while others are specific to mammalian mitochondria.

The mitochondrial transcription termination factor 4 (MTERF4) and the rRNA methyltransferase NSUN4 were among the first characterized mitoribosome assembly factors[4]. NSUN4 methylates C1488 in the 12S mt-rRNA of the small mitoribosomal subunit (mt-SSU)[5]. However, MTERF4 and NSUN4 also form a stoichiometric complex on the large mitoribosomal subunit (mt-LSU)[4,6,7], which is essential to facilitate monosome assembly[5].

During mt-LSU maturation, 16S mt-rRNA undergoes three methylations in the catalytically critical peptidyl transferase center (PTC). Human methyltransferase MRM1 is responsible for 2′-O-ribose methylation at G2815 of the P-loop[8,9], while U3039 and G3040, located in the A-loop of 16S mt-rRNA, are methylated by MRM2 and MRM3, respectively[8–10]. The importance of methylation for mitoribosomal function has recently been substantiated by the finding that mutations in MRM2 lead to a cellular respiratory deficiency and a clinically severe multisystemic disorder[11].

A growing number of studies in recent years have shown that a family of GTPBPs is crucial for mammalian mitoribosome assembly. MTG1 (GTPBP7), GTPBP6, and two homologs of bacterial ObgE, GTPBP10, and GTPBP5, assist late-stage maturation of the mt-LSU and their ablation leads to a severe translational defect[12–18].

Although assembly factors are central to mammalian mitoribosome biogenesis, their molecular mechanisms of action remain largely enigmatic. Indeed, while the assembly pathways of the mt-SSU and mt-LSU in trypanosomes have recently been visualized by cryo-EM[19–21], only one structural study of native human mt-LSU intermediates is available[22]. In these assembly intermediates, over a fifth of the total 16S mt-rRNA (which in mature mitoribosome comprises the PTC and intersubunit bridges) is not yet folded, while the MRP bL36m protein is missing[22]. How and when the PTC is formed and the role of the assembly factors in this process remain to be elucidated.

In this study, to provide further insights into the process of the mt-LSU assembly in human mitochondria, we determine the cryo-EM structures of mt-LSU intermediates isolated from GTPBP5-deficient cells and complement it with the analysis of a GTPBP5-bound mt-LSU complex immunoprecipitated from cells overexpressing a FLAG-tagged version of GTPBP5. The obtained structures uncover mechanisms of action for several late-stage mt-LSU assembly factors unique to mammalian mitochondria and reveal an unexpected role for mtEF-Tu in the process. These data represent visualizations of the conformational changes essential for the final stages of mt-rRNA folding and reveal the molecular basis for a critical sequence of events leading to mature mt-LSU formation in humans.

## Results

**Composition of the GTPBP5^KO and GTPBP5^IP mt-LSU assembly intermediates.** We purified mitoribosomes from a genetically engineered human cell line that lacks GTPBP5

(GTPBP5^KO) and from a cell line expressing 3xFLAG-tagged GTPBP5 (GTPBP5^IP) (Supplementary Fig. 1a, b), and used single-particle cryo-EM to determine their structures. Both the GTPBP5^KO and GTPBP5^IP mt-LSU assembly intermediates reveal several trapped assembly factors: the MTERF4-NSUN4 complex, MRM2, MTG1, and the MALSU1:L0R8F8:mt-ACP module (Fig. 1a, b, Supplementary Figs. 2–4). Furthermore, the GTPBP5^IP mt-LSU structure features GTPBP5 and the mitochondrial elongation factor mtEF-Tu (Fig. 1b, Supplementary Fig. 3). Comparing the GTPBP5^KO and the GTPBP5^IP mt-LSU intermediates with the mature mt-LSU[23] reveals two crucial differences in the 16S rRNA conformation (Fig. 1c, d). First, in both GTPBP5^KO and GTPBP5^IP intermediates, MTERF4 in the MTERF4-NSUN4 complex, binds an immaturely folded region of the 16S domain IV. This region (C2548-G2631)—corresponding to helices H68, H69, and H71 of the mature mt-LSU—is folded into an intermediate rRNA helical structure (pre-H68-71) (Fig. 1). The pre-H68-71 occupies a different position on the mt-LSU than H68-71 in the mature mt-LSU, where helices H68-71 and H89-90 jointly form the PTC (Fig. 1d). MTERF4 binding of pre-H68-71 partly orders the disordered rRNA in the mt-LSU assemblies' subunit interface side and thereby enables MRM2 to bind (Fig. 1d). Second, in the GTPBP5^KO—but not in the GTPBP5^IP—the junction between H89 and H90 of domain V is significantly different compared to the mature mitoribosome (Fig. 1c, d). Specifically, at the base of H89, one helical turn remains unfolded and instead forms a loop (Fig. 1d).

**MTERF4–NSUN4 complex steers the final steps of 16S mt-rRNA folding and allows for MRM2 binding.** The MTERF4–NSUN4 complex, previously shown to be essential for monosome assembly[4,5], binds at the intersubunit interface in the GTPBP5^KO and GTPBP5^IP structures (Fig. 1a, b). The C-terminal part of MTERF4 binds to NSUN4 close to the NSUN4 N-terminus in a mixed hydrophobic-polar binding interface, similar to earlier crystal structures of the isolated complex[6,7] (Fig. 2a). NSUN4 was previously shown to m5C-methylate the C1488 in 12S mt-rRNA[5]. In our structures, the active site of NSUN4 is turned toward the mt-LSU core (Supplementary Fig. 5a), impeding methylation of the 12S mt-rRNA. Although the methyl-donor S-adenosyl-methionine (SAM) is observed in the NSUN4 active site, no RNA substrate is present. Furthermore, in the GTPBP5^KO and GTPBP5^IP structures, the MTERF4–NSUN4 complex is bound and bent from two sides by uL2m. Specifically, the uL2m C-terminus penetrates in between NSUN4 and MTERF4 to further stabilize the MTERF4–NSUN4 binding interface and decrease the curvature of the MTERF4 solenoid relative to the crystal structures (Fig. 2a). This reforming of the MTERF4 solenoid is necessary to bind the pre-H68-71 rRNA region in the strongly positively charged concave side of MTERF4 (Supplementary Fig. 5b). Here, MTERF4 forms an extensive network of contacts with the pre-H68-71 that stabilizes the association and promotes formation of the H68-71 premature fold (Fig. 2b; Supplementary Fig. 5c). The mature H71 base-pairing is already formed within the pre-H68-71. Thus, MTERF4 initiates the folding of this 16S mt-rRNA region. Furthermore, it also exposes the A-loop, which is obstructed by H68, H69, and H71 in the mature mt-LSU (Fig. 1d), thereby allowing MRM2 binding.

Similarly to a previously determined mt-LSU assembly intermediate[22], there is a MALSU1-module positioned adjacent to uL14m in both the GTPBP5^KO and GTPBP5^IP (Fig. 1a, b). Furthermore, MTG1 (GTPBP7), which assists in late-stage mt-LSU maturation[17], is bound in the vicinity of the pre-H68-71 (Fig. 1a, b). MTG1 contacts the C-terminus of MALSU1

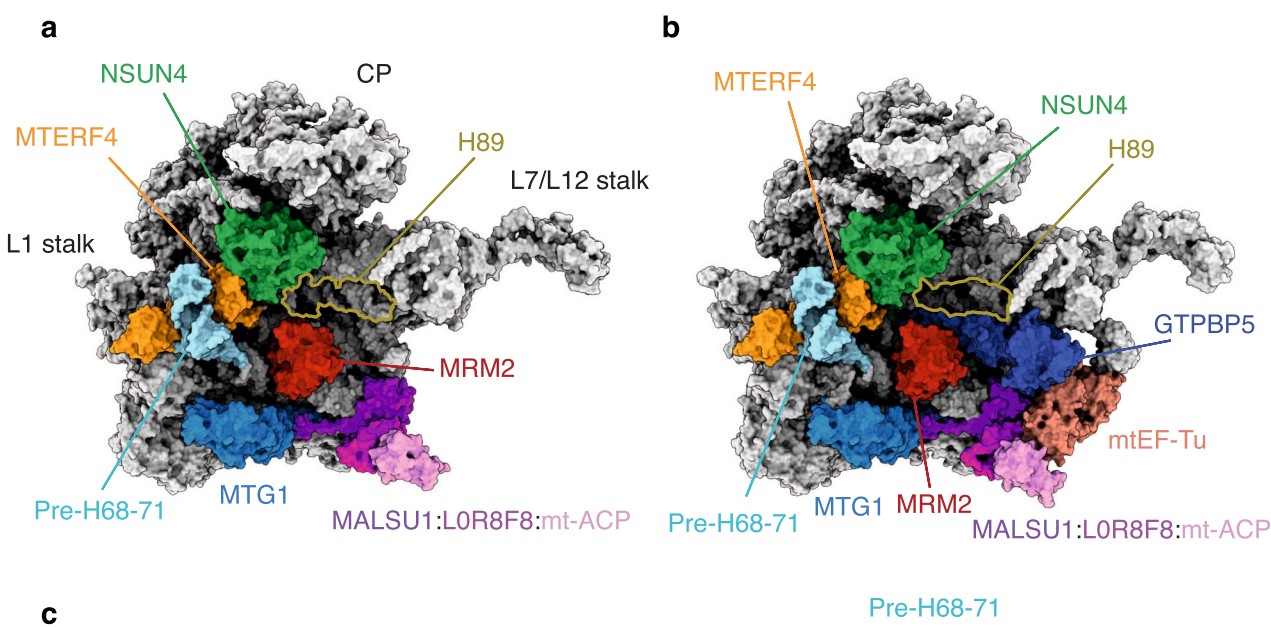

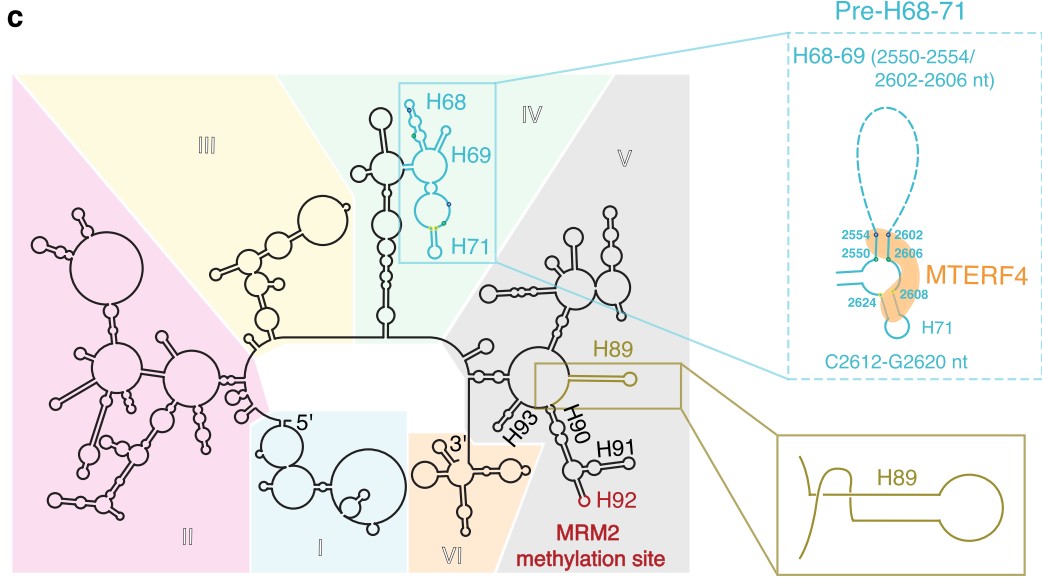

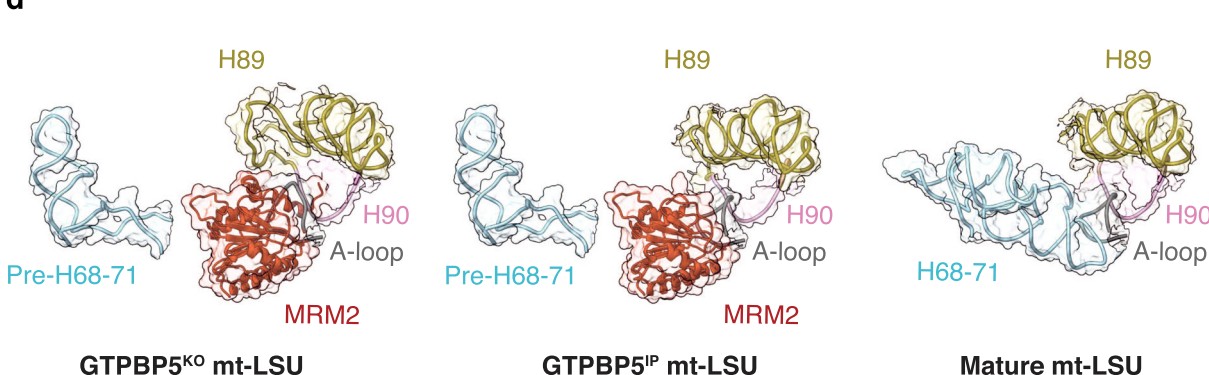

**GTPBP5$^{KO}$ mt-LSU**    **GTPBP5$^{IP}$ mt-LSU**    **Mature mt-LSU**

(Supplementary Fig. 6a) and the region encompassing A2554–U2602 of the pre-H68-71. This region could not be modeled due to the lower local resolution, but the contact is visible in the map (Supplementary Fig. 6b). Interestingly, the position of human MTG1 in our structures differs significantly from its bacterial and trypanosomal counterparts[19,21,24] (Supplementary Fig. 6c). Specifically, while in other systems MTG1

homologs contact the rRNA, reaching out toward the PTC (Supplementary Fig. 6c), in the trapped intermediates described here MTG1 is unlikely to induce pronounced conformational changes of the PTC.

MRM2 2′-O-methylates U3039 in the 16S A-loop during mt-LSU assembly[9,10] and in our GTPBP5$^{KO}$ and GTPBP5$^{IP}$ structures, MRM2 binds in the mt-LSU intersubunit interface

**Fig. 1 Overview of the GTPBP5[KO] and the GTPBP5[IP] mt-LSU assembly intermediates and comparison with the mature mt-LSU. a** Model-based surface representation of the GTPBP5[KO] structure bound by MTERF4, NSUN4, MRM2, MTG1, and the MALSU1 module. Mitoribosomal proteins and 16S mt-rRNA are shown in grey. The pre-H68-71 region bound to MTERF4 is highlighted as well as H89. **b** Model-based surface representation of the interface of the GTPBP5[IP] mt-LSU intermediate associated with MTERF4, NSUN4, MRM2, MTG1, MALSU1:L0R8F8:mt-ACP complex, GTPBP5 and mtEF-Tu. Pre-H68-71 bound to MTERF4 is shown in light blue. **c** Secondary structure of the mature mt-LSU 16S mt-rRNA. Differences in the rRNA fold of the GTPBP5[KO] mt-LSU intermediate are shown in the zoom-in views. Dashed lines indicate regions that are not modeled. Reference base-paired nucleotides of the pre-H68-71 are indicated with colored circles (blue, green, and yellow) both in the mature 16S mt-rRNA panel and in the pre-H68-71 zoom-in view. MRM2 methylation site (H92) is indicated in red. The six 16S mt-rRNA domains are shown in different colours. **d** Positioning of pre-H68-71, helices H89, H90 and the A-loop in GTPBP5[KO] mt-LSU (left), GTPBP5[IP] mt-LSU (middle), and the mature mt-LSU (right) (PDB:6ZSG[30]). In the GTPBP5[KO] and the GTPBP5[IP] mt-LSU structures, MRM2 is present. Model-based surface representation is shown.

(Fig. 1a, b). It features two N-terminal α-helices followed by a canonical S-adenosyl-L-methionine-dependent methyltransferase domain (SAM MTase) (Fig. 2c). In GTPBP5[KO], but not in GTPBP5[IP], the two N-terminal α-helices extend from MRM2 and insert into the rRNA core to thereby displace and retrieve the A-loop (16S mt-rRNA domain V) through a complex interaction network (Fig. 2d, e; Supplementary Fig. 7a). This places the 2′-hydroxyl of U3039 close to the ideal methyl-acceptor position in the MRM2 active site (Fig. 2d). However, there is no density for either SAM or S-adenosyl homocysteine (SAH) in the MRM2 active site and there is no apparent density for a 2′-O-methyl on U3039 (Supplementary Fig. 7b). Interestingly, G3040 that is 2′-O-methylated by MRM3[9,10], is methylated in our structures (Supplementary Fig. 7b). Hence, 2′-O-methylation by MRM3 takes place prior to MRM2 methylation in human mitoribosome biogenesis.

**GTPBP5 promotes remodeling of the PTC.** GTPBP5 consists of a glycine-rich N-terminal domain (Obg-domain) and a C-terminal GTPase domain (G-domain) (Fig. 3a). In our GTPBP5[IP] structure, the G-domain has GTP in its active site and is wedged between the L7/L12 stalk and the MALSU1 module (Fig. 1b and Supplementary Fig. 8a). The Obg-domain protrudes into the PTC (Fig. 3b), thereby displacing the A-loop from MRM2 active site and expelling the MRM2 N-terminal α-helices from the rRNA core (Fig. 3c), while the A-loop folds into the fully mature position (Fig. 3b3).

The protruding Obg-domain is positioned between H89 and H93 and occupies the space that accommodates the acceptor arm of the A-site tRNA during translation (Supplementary Fig. 8b). Hereby, GTPBP5 binds to the mt-LSU in a manner similar to the A-site tRNA, as it has been observed for ObgE in *Escherichia coli*[25]. The Obg-domain contains six glycine-rich sequence motifs that form antiparallel polyproline-II helices (helices a–f) (Fig. 3b). Helices c and d bind the A-loop, while the loop between helices e and f inserts into the major groove of H93. The loop between a and b inserts at the triple-junction formed between H89–H90–H93 (Fig. 3b).

Comparison of the GTPBP5[KO] and the GTPBP5[IP] structures with the mature mt-LSU reveals extensive maturation of the PTC upon GTPBP5 binding. The partly unfolded H89 in the GTPBP5[KO] is folded in the GTPBP5[IP] (Fig. 3b1). This folding is coordinated by the joint action of GTPBP5 and NSUN4: in the presence of GTPBP5, the extended N-terminal region of NSUN4 inserts into the rRNA core and temporarily displaces the P-loop (Fig. 3b2), thereby breaking the P-loop interaction with H89 (Fig. 3d, Supplementary Fig. 8c). As a consequence, H89 is given the space necessary to fold into a structure similar to its mature form (Fig. 3b1, lower).

The GTPBP5[IP] structure shows a rotation of the L7/L12 stalk in comparison to the GTPBP5[KO] structure (Supplementary Fig. 8d). Here, A2178 in H43 and U2205 in H44 of the 16S mt-rRNA in the L54/L11 region of the stalk form π-stacking interactions with two residues of the GTPase switch I element of GTPBP5 (Supplementary Fig. 8a, d). Thereby, the L7/L12 stalk stabilizes the "state 2" conformation of switch I. In this way, the rotated L7/L12 stalk stabilizes the GTP-state of GTPBP5[26] and consequently, a GTP is bound in our structure (Supplementary Fig. 8a, d). The requirement for GTPBP5 to be in a GTP-bound state is supported by the inability of a GTPBP5 Walker A mutant (GTPBP5-S238A) to bind mt-LSU intermediates[16]. A back-rotation of the L7/12 stalk, presumably by binding of another maturation factor to the mt-LSU assembly intermediate, would lead to a release of the switch I and the activation of GTP hydrolysis, followed by the release of GTPBP5 from the mt-LSU assembly intermediate. Taken together, GTPBP5 plays a direct and active role in rRNA remodeling and, together with the NSUN4 N-terminus, orchestrates the maturation of mitoribosomal PTC.

**Translation elongation factor mtEF-Tu is involved in mitoribosome assembly.** In the GTPBP5[IP] dataset, mtEF-Tu is present in approximately 75% of the GTPBP5-containing pre-ribosomal particles. This constitutes the structural evidence for binding of mtEF-Tu to mitoribosome assembly intermediates during biogenesis and confirms earlier mass-spectrometry data on isolated GTPBP5[IP] assembly intermediates as well as protein-proximity interactome analysis[16,27].

mtEF-Tu consists of a GTPase domain (G-domain; domain I) and two structurally similar β-stranded domains (domains II and III) (Fig. 4a). It was recently shown that during translation, mtEF-Tu·GTP delivers aminoacylated-tRNA to the mitoribosome in a manner similar to its bacterial EF-Tu counterparts (Supplementary Fig. 9a[28]). In contrast, the binding of an EF-Tu·GTP·aa-tRNA complex is sterically hindered by the MALSU1 module bound in our mt-LSU intermediates (Supplementary Fig. 9a). Unexpectedly, mtEF-Tu binds to the mitoribosome in a unique manner in the GTPBP5[IP] structure (Fig. 4b). Here, domains II and III establish extensive interactions with GTPBP5, the sarcin-ricin loop (SRL), and the MALSU1 stalk (Fig. 4b, c; Supplementary Fig. 9b). In addition, the G-domain switch I element, in its "state 1"/GDP conformation (Supplementary Fig. 9c), extends and binds MALSU1 (Fig. 4c).

The G-domain of mtEF-Tu does not contact the SRL as in mtEF-Tu's canonical role in translation but instead binds to the C-terminal region of one copy of bL12m[29,30] that also contacts uL10m at the stalk base (Fig. 4b). In bacteria, homologs to bL12m and uL10m recruit and activate translational GTPases such as EF-Tu via the bL12m C-terminal domain[31,32] and stimulate GTP hydrolysis[33]. Taken together, this suggests that mtEF-Tu hydrolysis—stimulated by bL12m and uL10m—is used to accommodate GTPBP5 on the maturating mt-LSU in analogy to the canonical EF-Tu role in translation, in which aminoacylated-tRNA is accommodated on the translating ribosome (Supplementary Fig. 9a).

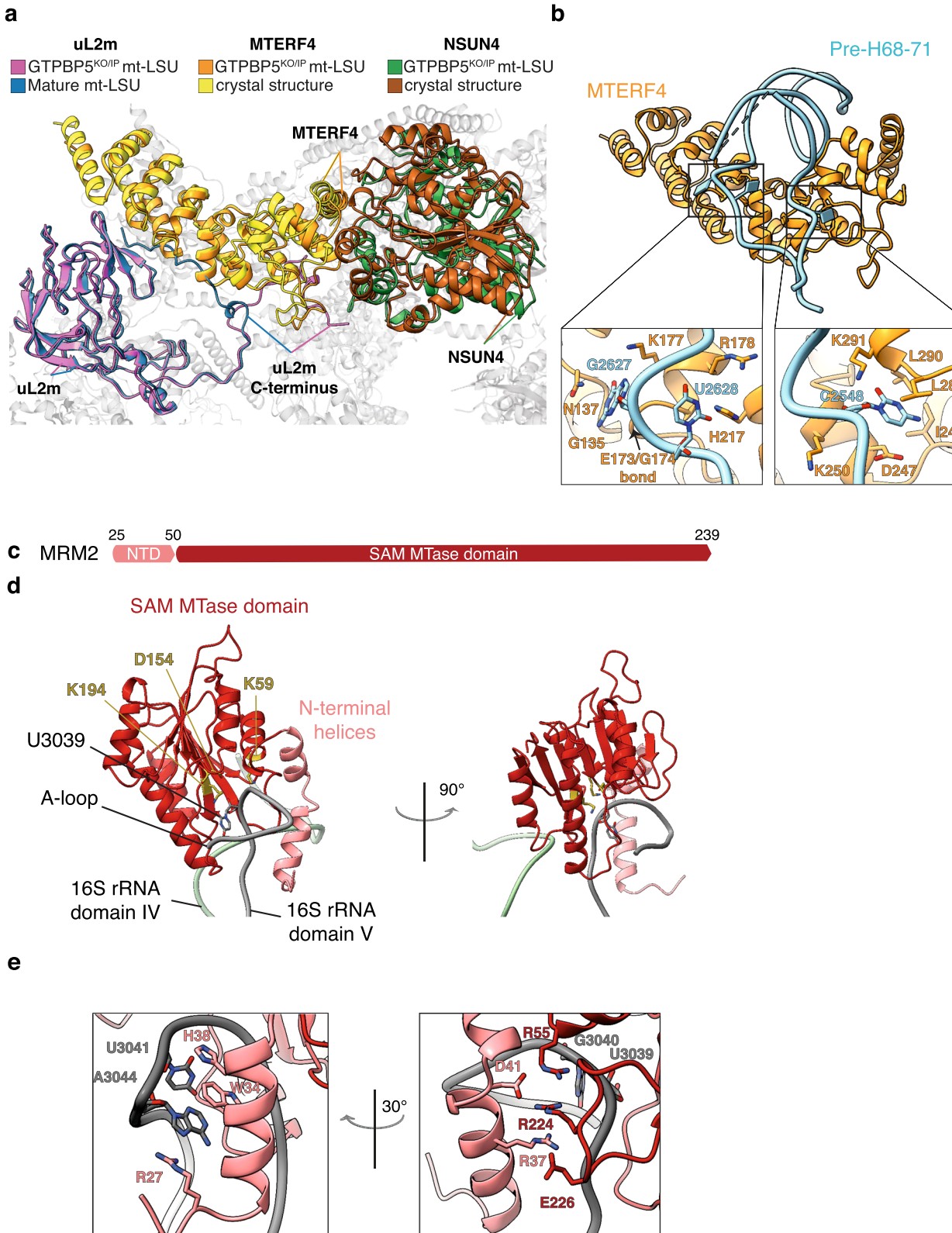

**Fig. 2 MTERF4–NSUN4 and MRM2 interaction with the mt-LSU assembly intermediates. a** Comparison of the MTERF4–NSUN4 complex bound to the GTPBP5[KO/IP] mt-LSU (orange and green, respectively) with the MTERF4–NSUN4 crystal structure (PDB: 4FP9[6]) (yellow and brown, respectively), and of uL2m from the GTPBP5[KO/IP] mt-LSU (pink) with uL2m from the mature mt-LSU (blue) (PDB: 3J7Y[23]). The uL2m C-terminus is indicated in both structures. Pre-H68-71 is not shown. **b** MTERF4–NSUN4 complex bound to pre-H68-71. Zoom-in panels show the interactions of MTERF4 with the pre-H68-71. **c** Schematic representation of MRM2 domains (NTD—light pink, SAM MTase domain—red). **d** MRM2 interaction with the domain IV rRNA (nucleotides 2644–2652, green) and the A-loop (grey). The MRM2 methylation site (U3039), as well as the catalytic triad of MRM2 (K59, D154, and K194), are highlighted as sticks. **e** Zoom-in views showing MRM2 interactions with the A-loop in different orientations.

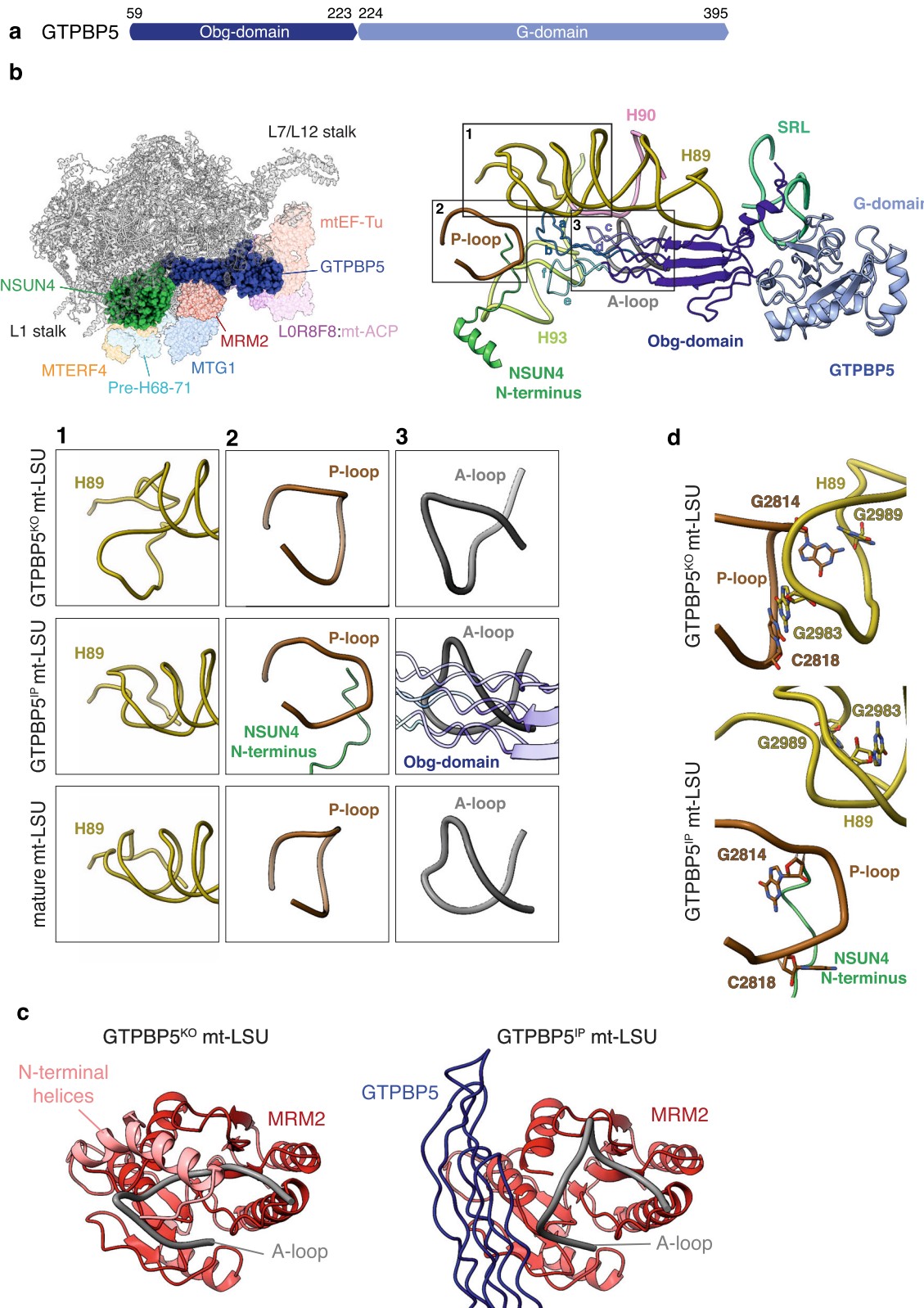

## Discussion

We isolated mitoribosomes from a cell line with knocked-out GTPBP5 and determined the single-particle cryo-EM structure of the purified mt-LSU assembly intermediates (Supplementary Fig. 1a). This approach allows the capture and molecular structure determination of normally short-lived assembly intermediates that do not accumulate in large numbers under physiological conditions. In order to trap sequential steps of mammalian mitoribosome biogenesis, we also determined the molecular structure of a GTPBP5[IP] complex immunoprecipitated from cells expressing a FLAG-tagged version of GTPBP5 (Supplementary Fig. 1b). The advantage of the immunoprecipitation strategy over in vitro addition of the factor to purified complexes relies on the fact that the natively expressed protein is able to bind

**Fig. 3 GTPBP5 contributes to the maturation of the PTC region. a** Schematic representation of GTPBP5 domains (Obg-domain dark blue, G-domain light blue). **b** Overview of GTPBP5 interactions with the 16S mt-rRNA (left panel) and corresponding zoom-in panel (right panel). The Obg-domain (dark blue) contacts helices that are in the PTC region: P-loop, A-loop, H89, H90, H93. Helices a–f of GTPBP5 Obg-domain are indicated. The SRL and the NSUN4 N-terminus are shown. Boxes 1–3 show the remodeling of the PTC in GTPBP5[IP] mt-LSU (middle panel) compared with GTPBP5[KO] mt-LSU (upper panel) and with the mature mt-LSU (lower panel) (PDB: 6ZSG[30]). **c** Comparison of MRM2 (red) and the A-loop (grey) conformations between GTPBP5[KO] mt-LSU (left) and GTPBP5[IP] mt-LSU (right). The N-terminal helices (pink) of MRM2 could not be modeled in the GTPBP5[IP] mt-LSU. The GTPBP5 Obg-domain is shown in dark blue. **d** Comparison of the P-loop and H89 conformations between GTPBP5[IP] mt-LSU (lower panel) and GTPBP5[KO] mt-LSU structures (higher panel).

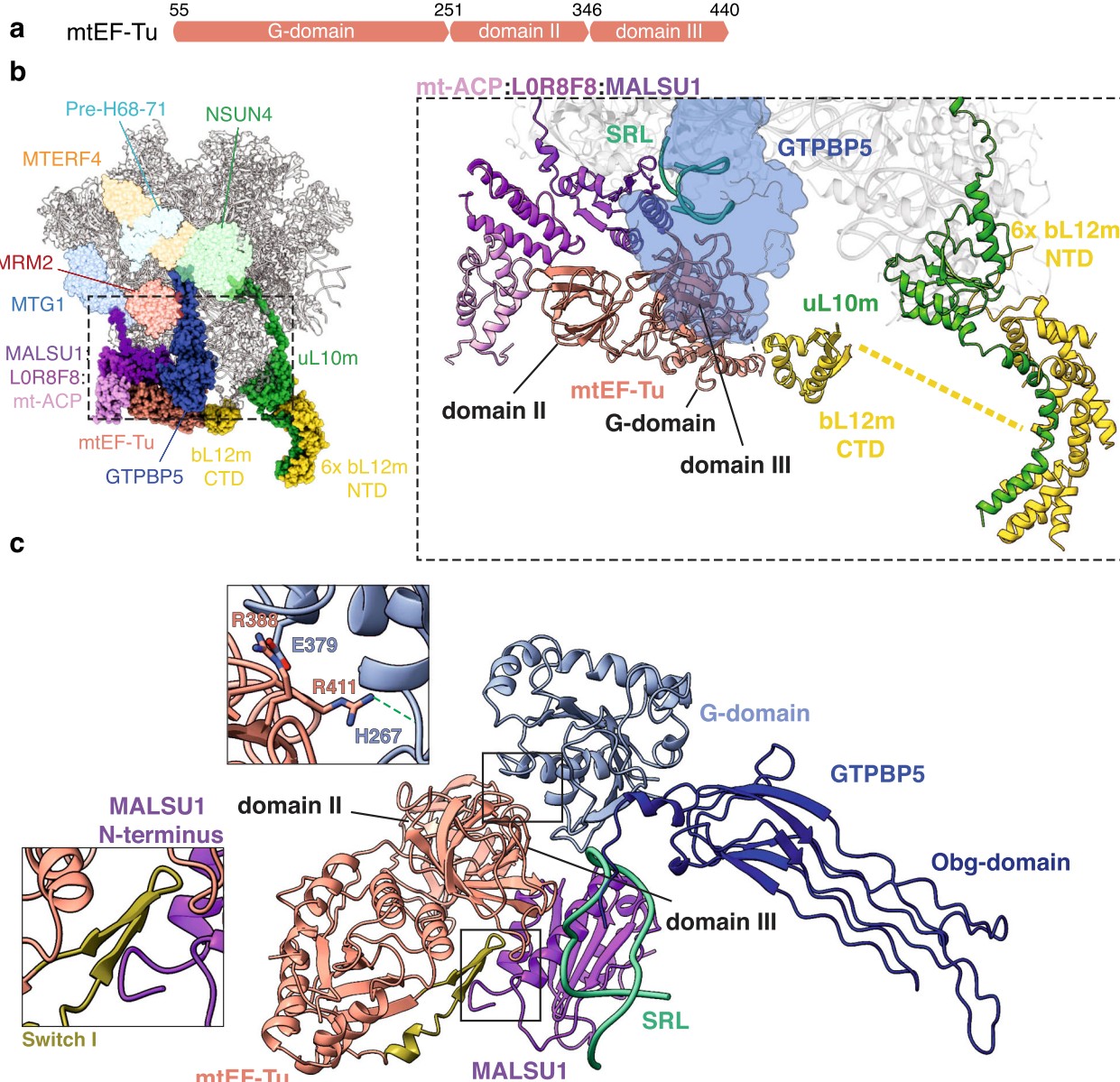

**Fig. 4 Interaction of mtEF-Tu with the mt-LSU assembly intermediate. a** Schematic representation of mtEF-Tu domains. **b** Overview of mtEF-Tu interaction with GTPBP5, the MALSU1 module, and the bL12m C-terminal domain (left panel) and corresponding zoom-in panel (right panel). mtEF-Tu G-domain, domain II and domain III and the SRL are indicated. The six copies of bL12m N-terminal domain[29,30] and uL10m are also highlighted. The yellow dashed line indicates a hypothetical connection between bL12m CTD and one of the six copies of bL12m NTD, not visible in the structure. **c** Representation of the mtEF-Tu interaction with GTPBP5 and MALSU1. The upper zoomed-in panel features interactions between the GTPBP5 G-domain and the mtEF-Tu domain III. The green dashed line indicates interactions to the RNA phosphate backbone. The lower zoom-in panel shows the mtEF-Tu switch I interaction with MALSU1.

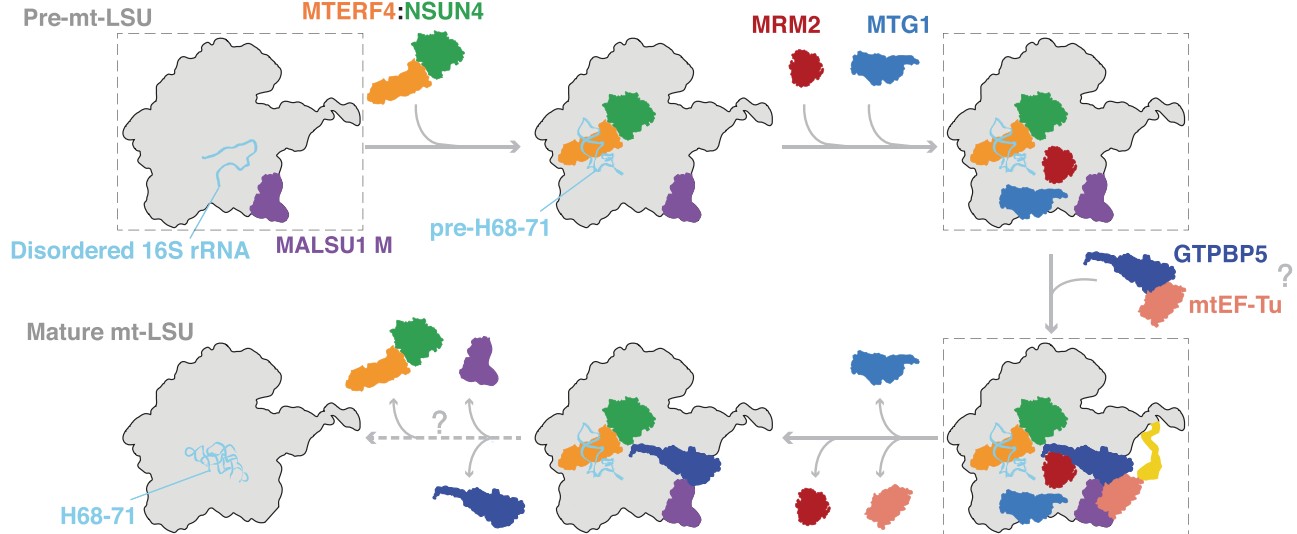

**Fig. 5 Model of the final steps of mt-LSU biogenesis.** Final steps of mt-LSU assembly. The steps representing resolved structures are highlighted in dashed boxes. The dashed arrow indicates that biogenesis factors are released in an unknown order. Question marks refer to not yet known mechanisms.

to its endogenous interactors in native states. Indeed, thanks to this method, we were able to detect the interaction of GTPBP5 and mtEF-Tu. Importantly, the striking similarities between the GTPBP5[KO] and GTPBP5[IP] structures suggest that GTPBP5[KO] likely represents a true assembly stage, not an "off-pathway" intermediate accumulated due to the absence of GTPBP5.

Thanks to our approaches, we were able to uncover features unique to mammalian mitochondria and explain the essential roles of several assembly factors that together promote fine RNA rearrangements and lead to the mt-LSU completion. Based on our structural data, on the different subpopulations obtained by our analysis and on previous biochemical studies, we propose a model of the late-stage mt-LSU assembly that requires the interplay of nine auxiliary factors (Fig. 5). The first intermediate of assembly proposed in our model corresponds to an mt-LSU intermediate population obtained from the GTPBP5[KO] dataset that displays a weak density at the intersubunit interface, corresponding to unstructured 16S mt-rRNA, and the associated MALSU1:L0R8F8: mt-ACP module, resembling the mt-LSU intermediate previously identified by Brown et al.[22] (Fig. 5; Supplementary Fig. 2a). This stage of assembly is followed by recruitment of the MTERF4–NSUN4 complex to the mt-LSU interface that leads to the folding of the pre-H68-71 helical structure by MTERF4, at the same time exposing the A-loop for MRM2 binding.

As a consequence of MTERF4–NSUN4 recruitment, MRM2 and MTG1 are engaged at the mt-LSU interface (Fig. 5), where MRM2 positions at the A-loop (Fig. 2d) and MTG1 contacts the tip of the pre-H68-71 (Supplementary Fig. 6b), which is in a strikingly divergent position compared to MTG1 homologs (Supplementary Fig. 6c). In bacteria, the MTG1 homolog RbgA binds the LSU in a similar location as MRM2 in our structures, contacting the 23S rRNA at the P- and A-site, and contributing to their maturation[24]. Likewise, Mtg1 in *Trypanosoma brucei* is bound in complex with the other two GTPases to 12S rRNA in a position similar to the one observed for RbgA[19,21]. Based on our structural data, it appears that human MTG1 does not induce pronounced conformational changes of the PTC or participate in the recruitment/dissociation of assembly factors, at the assembly stages captured in this study. MTG1 may instead function as a quality-control checkpoint by monitoring the folding status of H68-71 and preventing early subunit joining. Noteworthy, recent work by Chandrasekaran et al.[34] shows the structure of an intermediate of mt-LSU assembly with MTG1 bound in the same

location as MRM2 in our structure, suggesting that MTG1 changes its location on the mt-LSU during the assembly process.

The next step in our model consists in GTPBP5 arrival to the mt-LSU interface (Fig. 5). In particular, we show how GTPBP5 binding pushes the A-loop away from the MRM2 active site and at the same time how it facilitates the maturation steps of the PTC region (Fig. 3b, c). In addition, NSUN4 plays a direct part in PTC formation via its N-terminal extension and in coordination with GTPBP5 (Fig. 3b). Since the NSUN4 active site faces the mt-LSU core (Supplementary Fig. 5a), our structures strongly suggest two independent functions for NSUN4 in mitoribosome biogenesis: maturation of the PTC in mt-LSU assembly as shown in this work and methylation of C1488 in mt-SSU[5]. Whether 12S methylation activity occurs earlier or later in the assembly pathway or a separate pool of NSUN4 is responsible for this function, remains to be determined.

The arrival of GTPBP5 to the mt-LSU interface coincides with mtEF-Tu binding. mtEF-Tu is a highly conserved translational factor that uses its GTP-hydrolysis to accommodate aa-tRNAs to the A-site of the ribosome. In our structure, mtEF-Tu employs domain III to bind GTPBP5, which in turn protrudes into the PTC region, analogously to a tRNA during translation (Fig. 4b and Supplementary Fig. 8b). It is thus tempting to speculate that mtEF-Tu brings GTPBP5 to the mitoribosome in analogy to how it brings aa-tRNA during translation (Supplementary Fig. 9a). A recent analysis of the protein proximity network in mitochondria also confirmed a strong interaction of GTPBP5 and mtEF-Tu[27], however, further functional studies of this interaction are required to assess its physiological relevance.

Interestingly, we were able to visualize a direct interaction of mtEF-Tu with the C-terminal part of ribosomal protein bL12m (Fig. 4b). In bacteria, the L7/L12 stalk is necessary for optimal ribosomal translation[35] and a highly conserved C-terminal domain of L12 is responsible for an about 1000-fold stimulation of GTP hydrolysis by EF-Tu and EF-G[33]. Noteworthy, as previously observed in bacterial system[33], human bL12m also contacts uL10m at the stalk base. We speculate that mtEF-Tu GTPase activity is triggered by the interaction with the bL12m C-terminal domains, which is further promoted by their binding to uL10m.

Based on our data and on the subpopulations identified in our datasets, we hypothesize that the last steps of this model include the release of MTG1, followed by the release of the GDP-bound

mtEF-Tu, as well as MRM2. Later, GTPBP5, the MALSU1 module and MTERF4-NSUN4 are released in an unknown order and in a yet-to-be-defined modality. Importantly, while our manuscript was in revision, other studies[34,36–38] have shown different late steps of mt-LSU assembly involving the MTERF4-NSUN4 complex and other factors. In particular, in Hillen et al.[36], GTPBP6 is proposed to act at a later stage of assembly where MTERF4–NSUN4 and GTPBP5 are released, suggesting that GTPBP6 might be important for the release of these factors. Nevertheless, further studies are needed to completely elucidate these final steps of mt-LSU assembly.

In conclusion, our approaches provide a detailed view of the final stages of the mt-LSU biogenesis pathway. Moreover, as defects in mitoribosome biogenesis—resulting from, for example, mutations in MRM2 and GTPBP5—are increasingly implicated in mitochondrial disease[11,39], the current work does not only describe fundamental cellular processes but may also further diagnostic and therapeutic approaches to mitochondrial diseases.

## Methods

**Generation of GTPBP5 knock-out cell line**. The knock-out cell line (GTPBP5[KO]) was generated in the Flp-In T-Rex human embryonic kidney 293 (HEK293T) cell line (Invitrogen) using CRISPR/Cas9 technology targeted on exon 1 of *GTPBP5* gene, which encodes for GTPBP5, as described in Cipullo et al.[16]. In short, two pairs of gRNAs were designed and cloned into the pSpCas9(BB)-2A-Puro (pX459) V2.0 vector to generate out-of-frame deletions. Transfection of HEK293T cell line with the pX459 variants was performed using Lipofectamine 3000 following manufacturer's instructions. The selection of transfected cells was done using puromycin treatment at a final concentration of 1.5 mg/ml for 48 h. Subsequently, cells were single-cell diluted and transferred into a 96-well plate. Selected clones were screened via Sanger sequencing and Western blotting.

**Purification of the mt-LSU from GTPBP5[KO] cell line via sucrose gradient centrifugation**. Isolation of mitochondria was performed from eighty 150 mm dishes of GTPBP5[KO] cell line as described in Rorbach et al.[10], with some modifications. Crude mitochondria were further purified via differential centrifugation by being loaded onto a sucrose gradient (1 and 1.5 M sucrose, 20 mM Tris-HCl pH 7.5, 1 mM EDTA) and centrifuged at 107,170$g$ for 1 h at 4 °C (Beckman Coulter SW41-Ti rotor). Mitochondria forming a band at the interphase between the 1 and 1.5 M sucrose were collected and resuspended in 10 mM Tris-HCl pH = 7.5 in 1:1 ratio. After centrifugation, the final mitochondrial pellet was resuspended in mitochondrial freezing buffer (200 mM trehalose, 10 mM Tris-HCl pH 7.5, 10 mM KCl, 0.1% bovine serum albumin, 1 mM EDTA), snap frozen in liquid nitrogen and stored at −80 °C.

The mt-LSU was purified from the GTPBP5[KO] cell line via a sucrose gradient centrifugation experiment. Mitochondria were lysed at 4 °C for 20 min (25 mM HEPES-KOH pH = 7.5, 20 mM Mg(OAc)$_2$, 100 mM KCl, 2% (v/v) Triton X-100, 2 mM dithiothreitol (DTT), 1× cOmplete EDTA-free protease inhibitor cocktail (Roche), 40 U/μl RNase inhibitor (Invitrogen)) and later centrifuged at 15,871$g$ for 5 min at 4 °C. For mitoribosome purification, the mitolysate was subjected to sucrose cushion ultracentrifugation method (0.6 M sucrose, 25 mM HEPES-KOH pH = 7.5, 10 mM Mg(OAc)$_2$, 50 mM KCl, 0.5% (v/v) Triton X-100, 2 mM DTT) by being centrifuged at 913,774$g$ for 45 min at 4 °C (Beckman Coulter TL120.2 rotor). The mitoribosomal pellet was subsequently resuspended in ribosome resuspension buffer (25 mM HEPES-KOH pH = 7.5, 10 mM Mg(OAc)$_2$, 50 mM KCl, 0.05% DDM, 2 mM DTT) and centrifuged at 15,871$g$ for 10 min at 4 °C. The obtained supernatant was then loaded onto a linear sucrose gradient (15–30% (w/v)) in 1× gradient buffer (25 mM HEPES-KOH pH = 7.5, 10 mM Mg(OAc)$_2$, 50 mM KCl, 0.05% DDM, 2 mM DTT) and centrifuged for 2 h and 15 min at 93,500$g$ at 4 °C (Beckman Coulter TLS55 rotor). Fractions corresponding to the large mitochondrial subunit were collected and subjected to buffer exchange (25 mM HEPES-KOH pH = 7.5, 10 mM Mg(OAc)$_2$, 50 mM KCl) using Vivaspin 500 centrifugal concentrators.

**Generation of a mammalian cell line expressing GTPBP5**. A stable mammalian cell line overexpressing C-terminal FLAG-tagged GTPBP5 (GTPBP5::FLAG) in a doxycycline-inducible dose-dependent manner was generated as described in Cipullo et al.[16]. The GTPBP5 cDNA (hORFeome Database; Internal ID: 12579) was cloned into pcDNA5/FRT/TO. Flp-In T-Rex human embryonic kidney 293 (HEK293T, Invitrogen) cells were cultured in DMEM (Dulbecco's modified eagle medium) containing 10% (v/v) tetracycline-free fetal bovine serum, 2 mM Glutamax (Gibco), 1× Penicillin/Streptomycin (Gibco), 50 μg/ml uridine, 10 μg/ml Zeocin (Invitrogen), and 100 μg/ml blasticidin (Gibco) at 37 °C under 5% CO$_2$ atmosphere. Cells were seeded in a 6-well plate, grown in medium without antibiotics and co-transfected with pcDNA5/FRT/TO-GTPBP5::FLAG and pOG44

using Lipofectamine 3000 according to manufacturer's recommendations. After 48 h, the selection of cells was promoted by the addition of hygromycin (100 μg/ml, Invitrogen) and blasticidin (100 μg/ml) to culture media. After 2 to 3 weeks post-transfection, single colonies were picked and GTPBP5 overexpression was tested via Western Blot analysis 48 h after induction with 50 ng/ml doxycycline.

**Immunoprecipitation experiment**. Isolation and purification of mitochondria from 150 dishes (150 mm) of GTPBP5::FLAG overexpressing cell line were performed as described in the above paragraph "Purification of the mt-LSU from GTPBP5[KO] cell line via sucrose gradient centrifugation". The mt-LSU bound with GTPBP5 was isolated via FLAG-immunoprecipitation analysis (IP). Pelleted mitochondria were lysed at 4 °C for 20 min (25 mM HEPES-KOH pH = 7.5, 20 mM Mg(OAc)$_2$, 100 mM KCl, 2% (v/v) Triton X-100, 0.2 mM DTT, 1× cOmplete EDTA-free protease inhibitor cocktail (Roche), 40 U/μl RNase inhibitor (Invitrogen)) and centrifuged at 5000$g$ for 5 min at 4 °C. The supernatant was then added to ANTI-FLAG M2-Agarose Affinity Gel (Sigma-Aldrich) previously equilibrated (25 mM HEPES-KOH pH = 7.5, 5 mM Mg(OAc)$_2$, 100 mM KCl, 0.05% DDM) and incubated for 3 h at 4 °C. After incubation, the sample was centrifuged at 5000$g$ for 1 min at 4 °C, the supernatant was removed and the gel was washed three times with wash buffer. Elution (25 mM HEPES-KOH pH = 7.5, 5 mM Mg(OAc)$_2$, 100 mM KCl, 0.05% DDM, 2 mM DTT) was performed using 3× FLAG Peptide (Sigma-Aldrich) for about 40 min at 4 °C.

**Cryo-EM data acquisition and image processing**. Prior to cryo-EM grid preparation, grids were glow-discharged with 20 mA for 30 s using a PELCO easiGlow glow-discharge unit. Quantifoil Cu 300 mesh (R 2/2 geometry; Quantifoil Micro Tools GMBH) covered with a thin layer of 3 nm carbon were used for the GTPBP5[KO] sample. Carbon lacey films (400 mesh Cu grid; Agar Scientific) mounted with ultrathin carbon support film were used for the GTPBP5[IP] sample. Three μl aliquots of sample were applied to the grids, which were then vitrified in a Vitrobot Mk IV (Thermo Fisher Scientific) at 4 °C and 100% humidity (blot 3 s, blot force 3, 595 filter paper (Ted Pella Inc.)). Cryo-EM data collection (Supplementary Table 1) was performed with EPU (Thermo Fisher Scientific) using a Krios G3i transmission-electron microscope (Thermo Fisher Scientific) operated at 300 kV in the Karolinska Institutet's 3D-EM facility. Images were acquired in nanoprobe 165kX EFTEM SA mode with a slit width of 10 eV using a K3 Bioquantum during 1 s during which 60 movie frames were collected with a flux of 0.82 e−/Å$^2$ per frame. Motion correction, CTF-estimation, Fourier cropping (to 1.02 Å/px), picking and extraction in 600 pixel boxes (size threshold 300 Å, distance threshold 20 Å, using the pre-trained BoxNet2Mask_20180918 model) were performed on the fly using Warp[40]. Only particles from micrographs with an estimated resolution of 3.6 Å and under focus between 0.2 and 3 μm were retained for further processing.

For the GTPBP5[KO] dataset, 704,720 particles were picked from 37307 micrographs (Supplementary Fig. 2). The particles were imported into CryoSPARC 2.15[41] for further processing. After 2D classification, 130,289 particles were selected for an ab initio reconstruction. This reconstruction, in addition to two "bad" reconstructions created from bad 2D class averages, was used for heterogeneous refinement of the complete particle set resulting in one of the three classes yielding a large-subunit reconstruction with high resolution features (196,318 particles). After homogeneous refinement of these particles, the PDB model of a mitochondrial LSU assembly intermediate (PDB: 5OOL[22]) was fitted in the density. The reconstruction contained the MALSU1 module and also featured weak unexplained densities for several additional components in the intersubunit interface. A 3D variability analysis was performed with a mask on the intersubunit interface and a low pass resolution of 10 Å, and subsequently used for clustering into six particle classes representing different assembly intermediates. Two of the classes (43,057 and 41,619 particles) lacked the density for the A- and P-loops, H89, helices 68-71 and the L7/12 stalk. The A- and P-loops become visible in the third class (28,001 particles). The fourth class revealed a number of biogenesis factors: MRM2, MTERF4-NSUN4, MTG1, and the structured H67-H71 rRNA region (pre-H68-71) (48,646 particles) as well as H89. All the biogenesis factors are absent in the fifth class, in which helices 68 and 71 move to the mature position (26,678 particles). H69 is nevertheless not visible. The last class contains the assembled mitoribosome (8,317 particles). Non-uniform refinement of the fourth particle set yielded a reconstruction at 2.6 Å, which was used for model building and refinement. As the density for MTG1 was weaker than for the other factors, 3D variability analysis[42] was performed with a mask on the MTG1 region and with a 10 Å low-pass filter to select particles containing MTG1 (19,254 particles, which was subsequently subjected to homogeneous refinement yielding a reconstruction at 2.9 Å. The particles not containing MTG1 were refined to 2.8 Å.

For the GTPBP5[IP] dataset, 283,598 particles were picked from 112,076 micrographs using Warp and imported into CryoSPARC 2.15 (Supplementary Fig. 3)[41]. The complete particle set was used in heterogeneous refinement against the same three references derived from the GTPBP5[KO] dataset. One of the classes (78,306 particles) yielded a high-resolution reconstruction of the mt-LSU assembly intermediate. After homogeneous refinement, additional density for GTPBP5 was visible in the intersubunit interface. 3D variability analysis was performed with a mask on the GTPBP5 region and a low pass resolution of 10 Å. Subsequent clustering into two particle clusters revealed a particle subset containing GTPBP5

(71,834 particles), which was used for model building and refinement. This reconstruction also features densities for MRM2 and MTERF4–NSUN4. In addition, a weak density was present for mtEF-TU, the bL12m C-terminal domain, and MTG1. The refined particles were subjected to 2D classification and the bad classes were removed. The remaining particles were polished in RELION 3.1[43] and re-imported into CryoSPARC for further processing. Non-uniform refinement yielded a reconstruction at 3.1 Å, which was used for model building. 3D variability analysis was performed on these particles with a mask covering mtEF-TU, bL12m, and MTG1 and a 10 Å low pass filter. Subsequent clustering (4 clusters) revealed 2 clusters containing mtEF-TU/bL12m (17,886 particles in total), one cluster containing MTG1 (8,233 particles), and one cluster containing all three proteins (13,376 particles). The reconstructions derived from the MTG1- and the mtEF-Tu-containing particles reached a resolution of 3.2 Å.

**Model building and refinement**. Model building of the GTPBP5[KO] and GTPBP5[IP] mt-LSU assembly intermediate structures was performed using Coot[44]. The structure of a previous mt-LSU assembly intermediate (PDB 5OOL[22]) was used as a starting model. MRM2 and MTERF4–NSUN4 were identified by modeling secondary structure elements in Coot, and using the initial models for a structural search using the DALI server[45]. MTG1, as well as mtEF-Tu and the bL12m C-terminal domain in the GTPBP5-bound mt-LSU dataset, were identified using a density-based fold-recognition pipeline[22]. Using SWISS-MODEL[46], we generated homology models for the human GTPBP5 (template: PDB 4CSU chain 9[25]), MTG1 (template: PDB 3CNL chain A[47]), bL12m (template: PDB 1DD3 chain A[48]), and mtEF-TU (template: PDB 1D2E chain A[49]). All the models, as well as the crystal structure of the human MTERF4–NSUN4 (template: PDB 4FP9 chains A and B[6]), were fitted into the density map using Coot JiggleFit. The MTG1 GTPase domain and the L17/12 stalk were excluded from atomic refinement and were only subjected to rigid body refinement. Metal ions and modifications were placed based on map densities. Stereochemical refinement was performed using Refmac5[50]. Refinement statistics are reported in Supplementary Table 1, while modeled proteins and rRNA are shown in Supplementary Table 2. Validation of the final models was done via MolProbity[51]. Figures were generated using ChimeraX[52].

**Reporting summary**. Further information on research design is available in the Nature Research Reporting Summary linked to this article.

## Data availability

The coordinates and corresponding cryo-EM maps were deposited in the Protein Data Bank (PDB) and in the Electron Microscopy Data Bank (EMDB), respectively. For the GTPBP5KO mt-LSU assembly intermediate, the accession codes are 7O9M (atomic coordinates), EMD-12764 (main refinement map), EMD-12769 (particles with full MTG1 occupancy), and EMD-12770 (particles without MTG1). For the GTPBP5IP mt-LSU assembly intermediate, the accession codes are 7O9K (atomic coordinates), EMD-12763 (main refinement map), EMD-12767 (particles with full MTG1 occupancy) and EMD-12768 (particles with full mtEF-Tu occupancy). All other data can be obtained from the corresponding authors upon reasonable request.

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

## Acknowledgements

All cryo-EM data used in this work were collected at the Karolinska Institutet's 3D-EM facility. The SciLifeLab cryo-EM facility (used for grid preparation and initial screening) is funded by the Knut and Alice Wallenberg, Family Erling Persson, and Kempe Foundations. We acknowledge the support of: Knut and Alice Wallenberg Foundation (KAW 2018.0080) to B.M.H. and J.R.; the Swedish Research Council (2018-3808) to B.M. H.; Karolinska Institutet and the Max Planck Institute to J.R.. J.R. is a Fellow of the Knut and Alice Wallenberg Foundation (WAF 2017).

## Author contributions

M.C. with A.K. and J.R. help performed the cell biology, biochemistry, and sample preparations; G.V.G. and B.M.H. performed the cryo-EM data collection; G.V.G. and B. M.H. processed the data and determined the structures; G.V.G. built and refined models with help from A.K. and B.M.H.; M.C. and G.V.G. wrote the paper with input from all authors; J.R. and B.M.H. oversaw the project and edited the paper.

## Funding

## Competing interests

The authors declare no competing interests.
