## [Peer Review File · Nature Communications]

REVIEWERS' COMMENTS

Reviewer #1 (Remarks to the Author):

Cipullo and colleagues report cryo-EM structures of assembling mitoribosomes isolated from a human cell line either depleted of or overexpressing GTPBP5, a late-stage mitoribosomal assembly factor. Collectively, the structures show how nine assembly factors (MRM2, MTG1, mtEF-Tu, GTPBP5, the MTERF4:NSUN4 complex, and the MALSU1:LOR8F8:mt-ACP module) bind mitoribosomes and contribute to the folding and post-transcriptional modification of 16S rRNA. This is a high-quality structural paper that deserves publication.

I have few comments to make and do not think any additional experiments are necessary. However, I would recommend the authors take advantage of the more generous word count of Nature Communications to substantially expand the introduction and discussion, which are currently almost nonexistent. The introduction should cover what is already known about the assembly of mammalian mitoribosomes; identification of the factors involved and the current status of our structural understanding. What processes are upstream of GTPBP5? The discussion should try and link the study with the late stages of the assembly of the bacterial ribosome. What processes and assembly factors lie downstream of those identified here?

The discussion should also explore the possibility that the presence of mtEF-Tu is an artifact of overexpressing GTPBP5 and stalling a transient assembly complex rather than mtEF-Tu being a bona fide assembly factor. It was unclear why mtEF-Tu should bind specifically at this stage of the biogenesis pathway unless it was binding GTPBP5, which would be inconsistent with the implication on P6.L196 that mtEF-Tu, SRL, and MALSU1 form a platform for GTPBP5 binding.

I would also recommend not renaming H68-H71 as "helix-X". During assembly, many rRNA helices likely adopt transient structures that differ from the mature form. Renaming each of these has the potential to cause confusion. I suggest using "premature H68-H71" or equivalent.

For the GTPBP5KO dataset, all maps representing defined species should be deposited (for example maps lacking MTG1). Alternatively, or in addition, the full dataset should be uploaded to EMPIAR. EMDB AAAA and CCCC represent mixtures of species and should be clearly labeled as such when depositing to the EMDB.

Minor

1. Culture volumes should be provided in the Methods.
2. Resolutions should be reported to a single decimal place only.
3. P4.L129. Cryo-EM maps do not show "electron density".
4. P5.L159. GTPBP5 binding to a similar site as the tRNA acceptor arm does not constitute a "tRNA mimicry strategy".
5. P6.L172. "rRNA in the L54/L11 region" is confusing. Please define.
6. P9.L283, P9.L298 and P10.L309. Greek letter mu not displaying correctly.
7. What is the evidence that there are 6 copies of bL12m in human mitoribosomes?
8. Fig. 1 - it should be clarified that panels a and b show surface representations and not cryo-EM maps. Nucleotides should be added to the expansion in panel c to show which nucleotides are base-paired in premature H68-71.
9. Fig. 3 - the colors used for the loops for the obg domain in Figure 3b are too similar to the colors used for H89 and the P loop. This makes the figure very difficult to interpret.
10. Figs. 2-4 were easier to understand when simultaneously viewing Fig. 1A, suggesting they may benefit from a small panel providing positional context.
11. Ex. Data Fig. 3. - SAM should be shown in both rotations in panel a.
12. Ex. Data Fig. 5. - map contour levels should be provided for both panels.

Reviewer #2 (Remarks to the Author):

In this manuscript entitled "Structural basis for late maturation steps of the human mitoribosomal large subunit", Cipullo et al. describe cryoEM structures of two assembly intermediates of the mitoribosomal large subunit (LSU) isolated from human cells. To accomplish this, the authors used both KO and overexpression of a putative assembly factor GTPBP5 (GTPase), revealing two intermediates, denoted GTPBP5KO and GTPBP5IP, at resolutions between 2.65 and 3.21 Å respectively. These reconstructions represent late-stage intermediates of the assembly pathway in which the intersubunit region is decorated with a number of identified assembly factors that aid in the maturation of the peptidyl transferase center (PTC). Interestingly the authors discovered that mtEF-Tu is bound to the intermediate in the GTPBP5IP and based on the structure, the authors believe that it contributes to maturation of LSU, in addition to its characterized role in translation. Overall it is the opinion of this reviewer, that this manuscript should be published in Nature Communications, provided that the authors address the following points.

Major points:

1. A key concern in the ribosome assembly field is that depletion of key assembly factors can result in the accumulation of "off-pathway" intermediates. The authors should therefore at least state that the physiological relevance of the reconstruction originating from the knockout cell line is unclear as any observed conformational changes could occur simply due to the absence of GTPBP5.
2. The supplementary materials currently lack both local resolution analysis at the individual protein level as well as local density figures for all assembly factors. Since the overall resolution of the obtained particles is clearly dominated by the highly ordered solvent exposed regions of the maturing large ribosomal subunits, adding local resolution analyses of all assembly factors as well as showing the quality of the map used to identify and build each of the assembly factors will be essential. Similarly, for mtEF-Tu and GTPBP5 density figures will be required that ascertain the presence of GTP or GDP as shown in supplementary figure 6a,c and 7b
3. In supplementary figure 2, the authors have used 3D variability analysis in cryosparc to identify 3D classes that are "Mtg1-containing" or "mtEF-Tu containing". However, subsequently the authors have included an Mtg1 containing class within the mtEF-Tu containing pool, which seems counter-intuitive as the distinction between the resulting classes is the presence of either Mtg1 or mtEF-Tu. The pooling of four distinct states into two separate reconstructions (EMDB DDD or EMDB EEE) will therefore reduce the overall information that is available for the scientific community, and the authors should consider keeping and also depositing the four distinct classes that were obtained.
4. To clarify the proposed assembly pathway (Figure 4), the authors should indicate which of the visualized intermediates are based on the obtained reconstructions and which are based on additional biochemical data.
5. On page four the authors state that the observed position of Mtg1 makes it unlikely that this assembly factor can induce conformational changes. Therefore, the question arises: what is the role of human Mtg1 given its different binding site in the obtained reconstruction?
6. In cases where extensive sidechain and base interactions are shown (Figure 2b, d, e; Figure 3d; Figure 4c), corresponding density figures should be prepared to support the described level of detail.
7. Line 203-206: The authors make the argument that mtEF-Tu uses its GTPase activity to accommodate GTPBP5 on the maturing mtLSU. While this is an attractive model, the authors could be more cautious as there is no additional data besides the current cryo-EM reconstruction to support this model.
8. As three studies have now appeared on bioRxiv that describe human large subunit assembly intermediates, the authors may want to consider their data in light of the other data to put their findings into a broader context and to unify an assembly pathway.

Minor points:

1. For clarity, the authors should specify if density or model-based surface representation is used in Figure 1d.
2. Keeping one nomenclature consistently (Mtg2 or Gtpbp5) would greatly improve the clarity of the manuscript.
3. Line 122-123: Figure 1d does not clearly illustrate what is mentioned in the text as the A-loop is not indicated.
4. Figure 2a is hard to read, mostly since the authors have overlaid two structures. The clarity of this figure could be improved by removing the crystal structure and showing the overlay separately in the supplement. The same applies for supplementary figure 6c.
5. Figure 3b is very hard to read as three structures are compared with two color codes and the authors should consider revising this figure for clarity.
6. Figure 3d illustrates conformational changes of RNA but it is not clear if these occur due to the association of GTPBP5 (right panel) or as a result of its absence (left panel) – c.f. also major point 1 above.
7. Line 211: The authors claim that “approaches that combine biochemical tools with structure determination” have been employed in this study. However, only the cryo-EM structure is discussed so that the authors should reduce this claim.

Reviewer #3 (Remarks to the Author):

Many thanks for asking me to look at these two papers from the labs of Rorbach and Richter-Dennerlein. It was a pleasure. I'd really appreciate it if I could review the two of them together rather than try and dissect them or just repeat some of my comments for each of them. Further, I think it is really useful to review them together and commend you on asking reviewers to look at both together. I have absolutely no expertise in structural biology but I think I can comment more generally. Both pieces of work were of very high quality and address the pathway of mitoribosome biogenesis in human cell lines, particularly the maturation of the large subunit. The approach was similar in that Cipullo isolates partial complexes in the GTPBP5 KO cell line and Hellen in the GTPBP6 KO line. In addition, Cipullo use a GTPBP5 tagged IP and Hellen use an in vitro approach of adding back purified GTPBP6 to their partial complexes. There is substantial similarity between what is found by cryoEM with the GTPBP5 IP and the GTPBP6 KO but there are differences. For instance, in the IP Cipullo finds GTPBP7, MRM2 and intriguingly mt-EF-Tu. These are all not present in the cryoEM from the GTPBP6 KO. Why is this? Whilst intriguing, I am a little worried that Cipullo claim from these images that mtEF-Tu is involved in assembly of the mitoribosome. I think this is speculation and would need more supporting evidence from whole cell studies. There is strong evidence that GTPBP6 acts downstream of GTPBP5 and the in vitro studies, where purified GTPBP6 is added back and images of a more mature mtLSU with GTPBP6 bound and GTPBP5 + NSUN/MTERF4 absent does make one think that perhaps indeed GTPBP6 has somehow displaced this proteins. However, it is surely possible that GTPBP6 has bound to some intermediates in the preps that have lost these other components naturally and GTPBP6 has bound the free sites. I would be more convinced if a similar GTPBP6 IP had been performed from whole cells. However, it is entirely possible that GTPBP6 has indeed displaced these components.

Overall, I am impressed by the quality of both sets of data. There are some weaknesses and if the manuscripts can be toned down a little to indicate where weaknesses of their data interpretation may lie then I would be supportive of publication. One thing that intrigues me is that in the Cipullo paper there is no mention of GTPBP6. It is a ghost! Have Cipullo never come across this protein? Surely they must at least speculate that on the basis of their images it is very likely that other assembly

factors need to function further downstream of GTPBP5 to fully mature the LSU and that GTPBP6 could be one ? Then together these two manuscripts read very well and the GTPBP6 manuscript follows neatly on from the GTPBP5 work. Finally, I liked the final assembly figure of Cipullo and was not massively wowed by the video in Heller. Could I ask that a similar figure be included to help the reader in Heller et al. ?

Reviewer's Comments:

Reviewer #1 (Remarks to the Author)

Cipullo and colleagues report cryo-EM structures of assembling mitoribosomes isolated from a human cell line either depleted of or overexpressing GTPBP5, a late-stage mitoribosomal assembly factor. Collectively, the structures show how nine assembly factors (MRM2, MTG1, mtEF-Tu, GTPBP5, the MTERF4:NSUN4 complex, and the MALSU1:LOR8F8:mt-ACP module) bind mitoribosomes and contribute to the folding and post-transcriptional modification of 16S rRNA. This is a high-quality structural paper that deserves publication.

I have few comments to make and do not think any additional experiments are necessary. However, I would recommend the authors take advantage of the more generous word count of Nature Communications to substantially expand the introduction and discussion, which are currently almost nonexistent. The introduction should cover what is already known about the assembly of mammalian mitoribosomes; identification of the factors involved and the current status of our structural understanding. What processes are upstream of GTPBP5? The discussion should try and link the study with the late stages of the assembly of the bacterial ribosome. What processes and assembly factors lie downstream of those identified here?

We would like to thank the reviewer for the positive comments. As suggested, we have now considerably expanded the introduction and the discussion paragraphs according to the reviewer's suggestions.

The discussion should also explore the possibility that the presence of mtEF-Tu is an artifact of overexpressing GTPBP5 and stalling a transient assembly complex rather than mtEF-Tu being a bona fide assembly factor. It was unclear why mtEF-Tu should bind specifically at this stage of the biogenesis pathway unless it was binding GTPBP5, which would be inconsistent with the implication on P6.L196 that mtEF-Tu, SRL, and MALSU1 form a platform for GTPBP5 binding.

We understand the reviewer concerns regarding the overexpression of GTPBP5 and the possibility that GTPBP5-mtEF-Tu interaction might be an artefact. We were also concerned about it and further studies are needed to clarify the significance of this interaction. We will continue exploring this interesting observation, which we, however, believe is beyond the scope of this manuscript. For that reason, we have moderated our conclusions in the discussion (page 10, line 302-304):

"Recent analysis of the protein proximity network in mitochondria also confirmed a strong interaction of GTPBP5 and mtEF-Tu²⁷, however further functional studies of this interaction are required to assess its physiological relevance".

Furthermore, we agree with the reviewer's observation that the binding of mtEF-Tu to GTPBP5 would be inconsistent with mtEF-Tu, MALSU1 and the SRL working as a platform for GTPBP5 binding, therefore we deleted this sentence from the text.

I would also recommend not renaming H68-H71 as "helix-X". During assembly, many rRNA helices likely adopt transient structures that differ from the mature form. Renaming each of these has the potential to cause confusion. I suggest using "premature H68-H71" or equivalent.

We agree and replaced the term "helix-X" with "pre-H68-71" to refer to the premature conformation of H68-71 observed in our structures.

For the GTPBP5KO dataset, all maps representing defined species should be deposited (for example maps lacking MTG1). Alternatively, or in addition, the full dataset should be uploaded to EMPIAR. EMDB AAAA and CCCC represent mixtures of species and should be clearly labeled as such when depositing to the EMDB.

We added a map for the particles lacking MTG1 in the KO dataset (EMDB 12770 in the revised manuscript). Likewise, we described the different species in their respective EMDB entry.

Minor

1. Culture volumes should be provided in the Methods.

Thank you for spotting this omission. We have now provided the amounts of starting material used in our two approaches in the Methods section.

2. Resolutions should be reported to a single decimal place only.

Reported resolutions have now been rounded off to one single-decimal place only throughout the whole manuscript.

3. P4.L129. Cryo-EM maps do not show “electron density”.

Thank you for highlighting this inaccuracy. We have changed the sentence in which it occurred to “but the contact is visible in the map”.

4. P5.L159. GTPBP5 binding to a similar site as the tRNA acceptor arm does not constitute a “tRNA mimicry strategy”.

We agree with the reviewer’s comment. In the literature, ObgE (GTPBP5 bacterial homologue) binding to the 50S large subunit has been referred to as a ‘tRNA mimicry strategy’ (Feng et al., 2014 PMID: 24844575) and, since it binds in a notably similar way as GTPBP5, we decided to keep the same definition. Nevertheless, we agree that this conclusion is not fully accurate and we therefore modified the sentence as follows (page 6, line 191-192):

“Hereby, GTPBP5 binds to the mt-LSU in a manner similar to the A-site tRNA, as it has been observed for ObgE in *E. coli*²⁵”.

5. P6.L172. “rRNA in the L54/L11 region” is confusing. Please define.

Thank you for highlighting this imprecision. We have now clarified the specific nucleotides as follows (page 7, line 205-207):

“Here, A2178 in H43 and U2205 in H44 of the 16S mt-rRNA in the L54/L11 region of the stalk form π -stacking interactions with two residues of the GTPase switch I element of GTPBP5”.

6. P9.L283, P9.L298 and P10.L309. Greek letter mu not displaying correctly.

Thank you for noticing these typos. They have all been fixed now.

7. What is the evidence that there are 6 copies of bL12m in human mitoribosomes?

Previous computational studies predicted the presence of six copies of mitochondrial bL12m N-terminal domain (Davydov et al., 2013 PMID: 23340427). Moreover, recent work by Aibara et al. showed a well resolved L7/L12 stalk, which also evidenced the presence of six copies of bL12m N-terminal domain (Aibara et al., 2020 PMID: 32812867). We added these references to the manuscript (References 29 and 30. Page 8, line 236. Page 27, line 738).

8. Fig. 1 - it should be clarified that panels a and b show surface representations and not cryo-EM maps. Nucleotides should be added to the expansion in panel c to show which nucleotides are base-paired in premature H68-71.

Thank you for the suggestions. We implemented all the changes in the figure legend and also in Fig. 1c.

9. Fig. 3 - the colors used for the loops for the obg domain in Figure 3b are too similar to the colors used for H89 and the P loop. This makes the figure very difficult to interpret.

Thank you for this observation. We have now assigned different colors to the six glycine-rich sequence motifs of GTPBP5 Obg-domain.

10. Figs. 2-4 were easier to understand when simultaneously viewing Fig. 1A, suggesting they may benefit from a small panel providing positional context.

We have provided an overview to give positional context to Fig. 3b and Fig. 4b that would be otherwise difficult to read, as suggested by the reviewer. However, we think that Fig. 2 does not need an overview as the orientation of the figure does not differ much from the overview presented in Fig. 1.

11. Ex. Data Fig. 3. – SAM should be shown in both rotations in panel a.

Thank you for noticing, we have now added SAM to the left panel of Extended Data (now Fig. 5a).

12. Ex. Data Fig. 5. – map contour levels should be provided for both panels.

We have now provided this information in the figure legend of Extended Data Fig. 5 (Now Fig. 7b).

Reviewer #2 (Remarks to the Author)

In this manuscript entitled “Structural basis for late maturation steps of the human mitoribosomal large subunit”, Cipullo et al. describe cryoEM structures of two assembly intermediates of the mitoribosomal large subunit (LSU) isolated from human cells. To accomplish this, the authors used both KO and overexpression of a putative assembly factor GTPBP5 (GTPase), revealing two intermediates, denoted GTPBP5KO and GTPBP5IP, at resolutions between 2.65 and 3.21 Å respectively. These reconstructions represent late-stage intermediates of the assembly pathway in which the intersubunit region is decorated with a number of identified assembly factors that aid in the maturation of the peptidyl transferase center (PTC). Interestingly the authors discovered that mtEF-Tu is bound to the intermediate in the GTPBP5IP and based on the structure, the authors believe that it contributes to maturation of LSU, in addition to its characterized role in translation. Overall it is the opinion of this reviewer, that this manuscript should be published in Nature Communications, provided that the authors address the following points.

We would like to thank the reviewer for the positive comments.

Major points:

1. A key concern in the ribosome assembly field is that depletion of key assembly factors can result in the accumulation of “off-pathway” intermediates. The authors should therefore at least state that the physiological relevance of the reconstruction originating from the knockout cell line is unclear as any observed conformational changes could occur simply due to the absence of GTPBP5.

We understand the reviewer’s concern, but we would like to argue that the similarities observed between the GTPBP5^{KO} and GTPBP5^{IP} structures strongly suggest that we were able to capture consecutive steps of assembly and that the structure we analysed in our GTPBP5^{KO} sample is likely representative of a true assembly intermediate rather than an “off-pathway” intermediate. Moreover, very recent studies showed similar structures that were obtained from different cellular models, in which GTPBP5 was present, further suggesting that what we observe is not simply linked to the absence of GTPBP5 (Hillen et al., 2021 doi: <https://doi.org/10.1101/2021.03.17.435767>; Lenarcic et al., 2021 doi: <https://doi.org/10.1101/2021.03.29.437532>).

We have highlighted this aspect in the manuscript as follows (page 8, line 255-257):

“Importantly, the striking similarities between the GTPBP5^{KO} and GTPBP5^{IP} structures suggest that GTPBP5^{KO} likely represents a true assembly stage, not an “off pathway” intermediate accumulated due to the absence of GTPBP5.”

2. The supplementary materials currently lack both local resolution analysis at the individual protein level as well as local density figures for all assembly factors. Since the overall resolution of the obtained particles is clearly dominated by the highly ordered solvent exposed regions of the maturing large ribosomal subunits, adding local resolution analyses of all assembly factors as well as showing the quality of the map used to identify and build each of the assembly factors will be essential. Similarly, for mtEF-Tu and GTPBP5 density figures will be requires that ascertain the presence of GTP or GDP as shown in supplementary figure 6a,c and 7b.

We now indicate the local resolution in Extended Data Figure 4. The density for the nucleotides bound to GTPBP5 and mtEF-Tu is shown in Extended Data Fig. 8a and 9b, respectively.

3. In supplementary figure 2, the authors have used 3D variability analysis in cryosparc to identify 3D classes that are “Mtg1-containing” or “mtEF-Tu containing”. However, subsequently the authors have included an Mtg1 containing class within the mtEF-Tu containing pool, which seems counter-intuitive as the distinction between the resulting classes is the presence of either Mtg1 or mtEF-Tu. The pooling of four distinct states into two separate reconstructions (EMDB DDD or EMDB EEE) will therefore reduce the overall information that is available for the scientific community, and the authors should consider keeping and also depositing the four distinct classes that were obtained.

In this classification, we sought to separate classes with 100% occupancy of MTG1 or mtEF-Tu in order to improve the density for these factors, irrespectively of whether the other factor is present or not. Indeed, the classification results suggest that MTG1 and mtEF-Tu bind independently from each other. Aside from the presence/absence of these two factors, the classes are identical. To clarify this, we have re-labelled the classes in Extended Data Fig. 3 as MTG1-containing (EMDB 12767) or mtEF-TU-containing (EMDB 12768). These two merged classes were deposited alongside the EMDB 12763, which shows the highest quality for the large subunit but with only partial occupancy of mtEF-Tu and MTG1.

4. To clarify the proposed assembly pathway (Figure 4), the authors should indicate which of the visualized intermediates are based on the obtained reconstructions and which are based on additional biochemical data.

Thank you for this comment. We would like to point out that with “biochemical tools” we refer to the biochemical approaches we employed in order to purify assembly intermediates that were later subjected to structural studies. For clarity, we have now explained more extensively this aspect of our study by also adding a supplementary figure (Extended Data Fig. 1) with the representation of the workflow used to obtain GTPBP5^{KO} and GTPBP5^{IP} structure.

To emphasize which of the assembly steps discussed in our work are based on our structural analyses and which are hypotheses at this point, in Fig. 5 we have highlighted in dashed boxes the intermediates of assembly that are based on our analysis. Moreover, we have expanded the discussion to better explain the reasoning behind the assembly pathway. We would like to add that our hypotheses regarding the order of arrival or release of the assembly factors throughout these steps is based on the identification of subpopulations in our analysis as well as on the biological relevance we observed in our structures. As an example, it is clear that MRM2 gets recruited after the arrival of MTERF4-NSUN4 complex as its binding would not occur without the formation of the pre-H68-71 helical structure.

5. On page four the authors state that the observed position of Mtg1 makes it unlikely that this assembly factor can induce conformational changes. Therefore, the question arises: what is the role of human Mtg1 given its different binding site in the obtained reconstruction?

Thank you for this very interesting comment. Unfortunately, with our current data it is hard to know what is the exact role of MTG1, at least at the assembly stage we have captured. Our analysis show that MTG1 contacts the pre-H68-71 fold and therefore we hypothesized that MTG1 might work as a quality control checkpoint by verifying the formation of the pre-H68-71, at the same time preventing premature subunit joining, as it has been suggested by previous biochemical studies (Kim et al., 2018 PMID: 30085276). However, a recent work by Chandrasekaran et al. (Chandrasekaran et al., 2021 doi: <https://doi.org/10.1101/2021.03.19.436169>) shows that, in their mt-LSU intermediate of assembly, MTG1 occupies the same position occupied by MRM2 in our structure, and contacts the A-loop, suggesting that MTG1 changes its location on the mtLSU during assembly process. We have now added a more extensive discussion on MTG1 role in our manuscript (page 9, line 271-285):

“As a consequence of MTERF4-NSUN4 recruitment, MRM2 and MTG1 are engaged at the mt-LSU interface (Fig. 5), where MRM2 positions at the A-loop (Fig. 2d) and MTG1 contacts the tip of the pre-H68-71 (Extended Data Fig. 6b), which is a strikingly divergent position compared to MTG1 homologues (Extended Data Fig. 6c). In bacteria, the MTG1 homologue RbgA binds the LSU in a similar location as MRM2 in our structures, contacting the 23S rRNA at the P- and A- site, and contributing to their maturation²⁴. Likewise, Mtg1 in *Trypanosoma brucei* is bound in complex with other two GTPases to 12S rRNA in a position similar to the one observed for RbgA^{19,21}. Based on our structural data, it appears that human MTG1 does not induce pronounced conformational changes of the PTC or participate in the recruitment/dissociation of assembly factors, at the assembly stages captured in this study. MTG1 may instead function in a quality-control checkpoint by monitoring the folding status of H68-71 and preventing early subunit joining. Noteworthy, recent work by Chandrasekaran et al.³⁴ shows the structure of an intermediate of mt-LSU assembly with MTG1 bound in the same location as MRM2 in our structure, suggesting that MTG1 changes its location on the mt-LSU during the assembly process.”

6. In cases where extensive sidechain and base interactions are shown (Figure 2b, d, e; Figure 3d; Figure 4c), corresponding density figures should be prepared to support the described level of detail.

We have now added the density figures in the Extended Data Fig. 5c (related to Fig. 2b), 7a (related to Fig. 2e), 8c (related to Fig. 3d, for GTPBP5^{KO} structure), 8a, 9b (related to Fig. 4c) and 9c. We do not show the density for Fig. 2d as this figure intends to show the overall fold of MRM2 and its position with respect to the A-loop, but not the atomic details of the interaction, which are shown in Fig. 2e.

7. Line 203-206: The authors make the argument that mtEF-Tu uses its GTPase activity to accommodate GTPBP5 on the maturing mtLSU. While this is an attractive model, the authors could be more cautious as there is no additional data besides the current cryo-EM reconstruction to support this model.

We agree with the reviewer's comment. As replied to reviewer 1, we are planning to perform further work to elucidate the nature of GTPBP5-mtEF-Tu interaction.

We have now moderated our conclusions in the discussion (page 10, lines 302-304):

"Recent analysis of the protein proximity network in mitochondria also confirmed a strong interaction of GTPBP5 and mtEF-Tu ²⁷, however further functional studies of this interaction are required to assess its physiological relevance."

8. As three studies have now appeared on biorxiv that describe human large subunit assembly intermediates, the authors may want to consider their data in light of the other data to put their findings into a broader context and to unify an assembly pathway.

We have now expanded our discussion paragraph taking into consideration the recently published papers on biorxiv:

(page 9, line 282-285):

"Noteworthy, recent work by Chandrasekaran et al. ³⁴ shows the structure of an intermediate of mt-LSU assembly with MTG1 bound in the same location as MRM2 in our structure, suggesting that MTG1 changes its location on the mt-LSU during the assembly process."

(page 10, lines 315-320):

"While our manuscript was in revision, other studies ^{34,36-38} have shown different late steps of mt-LSU assembly involving the MTERF4-NSUN4 complex and other factors. In particular, in Hillen et al. ³⁶, GTPBP6 is proposed to act at a later stage of assembly where MTERF4-NSUN4 and GTPBP5 are released, suggesting that GTPBP6 might be important for the release of these factors."

Minor

points:

1. For clarity, the authors should specify if density or model-based surface representation is used in Figure 1d.

We have now specified this in the figure legend of Fig. 1d.

2. Keeping one nomenclature consistently (Mtg2 or Gtpbp5) would greatly improve the clarity of the manuscript.

We have corrected it by using GTPBP5 nomenclature throughout the manuscript.

3. Line 122-123: Figure 1d does not clearly illustrate what is mentioned in the text as the A-loop is not indicated.

We have now displayed the A-loop in Fig. 1d.

4. Figure 2a is hard to read, mostly since the authors have overlaid two structures. The clarity of this figure could be improved by removing the crystal structure and showing the overlay separately in the supplement. The same applies for supplementary figure 6c.

We understand the reviewer's point of view and we have tried to accomplish the task but we have concluded that showing the overlay separately for both Fig. 2a and Extended Data Fig. 6c makes it harder to read as the small changes in conformation would be hardly visible. Therefore, we applied a different, more intuitive, labeling of the figures that we hope will make it clearer for the reader to understand.

5. Figure 3b is very hard to read as three structures are compared with two color codes and the authors should consider revising this figure for clarity.

We have now changed the organisation of Fig. 3b to three different panels with the same color code, representative of GTPBP5^{KO} (upper panel), GTPBP5^{IP} (middle panel) and the mature mt-LSU structures (lower panel), respectively. We hope these chronologically ordered panels will be clearer for the reader to compare.

6. Figure 3d illustrates conformational changes of RNA but it is not clear if these occur due to the association of GTPBP5 (right panel) or as a result of its absence (left panel) – c.f. also major point 1 above.

As mentioned above, we believe that thanks to the similarities observed between the GTPBP5^{KO} and the GTPBP5^{IP} structures, the two assembly intermediates represent physiologically-relevant states of assembly. As an additional

confirmation, two recent studies published on biorxiv (Hillen et al., 2021 doi: <https://doi.org/10.1101/2021.03.17.435767>; Lenarcic et al., 2021 doi: <https://doi.org/10.1101/2021.03.29.437532>) show similar mt-LSU intermediates where GTPBP5, together with NSUN4 N-terminus, interfere with the bond between the P-loop and H89, in the exact same manner as we observed (Fig. 3d). In these two studies, two different cellular models are used and they do not involve either the depletion or the overexpression of GTPBP5.

7. Line 211: The authors claim that “approaches that combine biochemical tools with structure determination” have been employed in this study. However, only the cryo-EM structure is discussed so that the authors should reduce this claim.

As mentioned in the above reply, the “biochemical tools” we address in the manuscript are the biochemical approaches performed to isolate mt-LSU assembly intermediates from GTPBP5^{KO} and GTPBP5^{IP} cells (Extended Data Fig. 1). We have now explained these two strategies in the manuscript so that the reader does not confuse them with the previous biochemical data (page 8, line 246). We have also removed the sentence about ‘approaches that combine biochemical tools with structure determination’.

Reviewer #3 (Remarks to the Author)

Many thanks for asking me to look at these two papers from the labs of Rorbach and Richter-Dennerlein. It was a pleasure. I'd really appreciate it if I could review the two of them together rather than try and dissect them or just repeat some of my comments for each of them. Further, I think it is really useful to review them together and commend you on asking reviewers to look at both together. I have absolutely no expertise in structural biology but I think I can comment more generally. Both pieces of work were of very high quality and address the pathway of mitoribosome biogenesis in human cell lines, particularly the maturation of the large subunit. The approach was similar in that Cipullo isolates partial complexes in the GTPBP5 KO cell line and Hellen in the GTPBP6 KO line. In addition, Cipullo use a GTPBP5 tagged IP and Hellen use an in vitro approach of adding back purified GTPBP6 to their partial complexes. There is substantial similarity between what is found by cryoEM with the GTPBP5 IP and the GTPBP6 KO but there are differences. For instance, in the IP Cipullo finds GTPBP7, MRM2 and intriguingly mt-EF-Tu. These are all not present in the cryoEM from the GTPBP6 KO. Why is this ?

We would like to thank the reviewer for the positive comments regarding our work.

From our observations, we believe our GTPBP5^{IP} structure represents an earlier stage of mt-LSU assembly when compared to the mt-LSU intermediate captured by Hillen et al. in GTPBP6 KO cells. Previous mass spectrometry analysis on GTPBP5 pull-downs showed the association of GTPBP5 with MTG1, MRM2 and mtEF-Tu and a possible interplay between GTPBP5 and MRM2 was suggested (Cipullo et al., 2021 PMID: 33283228; Maiti et al., 2020 PMID: 32652011). We indeed observe that MRM2 methylation occurs in the presence of GTPBP5, which displaces the A-loop outside of MRM2 active site, probably leading to its release. In Hillen et al. methylation by MRM2 is present, suggesting they captured a step downstream of the same assembly pathway. Regarding MTG1, although our structural data is not enough to define the exact role of this protein at the assembly stage captured by us, we believe it might work as a quality control checkpoint by monitoring pre-H68-71 folding and preventing monosome formation. However, in light of the recent paper published on biorxiv by Chandrasekaran et al. (Chandrasekaran et al., 2021 doi: <https://doi.org/10.1101/2021.03.19.436169>), MTG1 might also be important for PTC maturation (detection of the correct MRM2 methylation), as it was found bound in the same location as MRM2 in our structure. Nevertheless, MTG1 seems to act at an earlier stage than the one captured by Hillen et al.. Regarding mtEF-Tu, we hypothesize that it is important for the recruitment of GTPBP5 during the mt-LSU assembly process and we speculate that its GTP hydrolysis is needed to accommodate GTPBP5 on the mt-LSU interface. In Hillen et al., mtEF-Tu is not visible as it might have already been released. We captured mtEF-Tu in its GDP state, suggesting it may be ‘on its way out’ from the mt-LSU and we detected it only because we trapped GTPBP5-bound complexes.

Whilst intriguing, I am a little worried that Cipullo claim from these images that mtEF-Tu is involved in assembly of the mitoribosome. I think this is speculation and would need more supporting evidence from whole cell studies.

We completely agree with the reviewer and we are eager to further study the potential role of mtEF-Tu in mitoribosome assembly. We will continue to study this pathway but we plan to defer this investigation to a future study. We have moderated our conclusions in the discussion (page 10, line 302-304):

“Recent analysis of the protein proximity network in mitochondria also confirmed a strong interaction of GTPBP5 and mtEF-Tu²⁷, however further functional studies of this interaction are required to assess its physiological relevance.”

There is strong evidence that GTPBP6 acts downstream of GTPBP5 and the in vitro studies, where purified GTPBP6 is added back and images of a more mature mtLSU with GTPBP6 bound and GTPBP5 + NSUN/MTERF4 absent does make one think that perhaps indeed GTPBP6 has somehow displaced this proteins. However, it is

surely possible that GTPBP6 has bound to some intermediates in the preps that have lost these other components naturally and GTPBP6 has bound the free sites. I would be more convinced if a similar GTPBP6 IP had been performed from whole cells. However, it is entirely possible that GTPBP6 has indeed displaced these components.

Overall, I am impressed by the quality of both sets of data. There are some weaknesses and if the manuscripts can be toned down a little to indicate where weaknesses of their data interpretation may lie then I would be supportive of publication.

Thank you for the nice comments. We have now expanded the discussion and moderated our conclusions.

One thing that intrigues me is that in the Cipullo paper there is no mention of GTPBP6. It is a ghost! Have Cipullo never come across this protein? Surely they must at least speculate that on the basis of their images it is very likely that other assembly factors need to function further downstream of GTPBP5 to fully mature the LSU and that GTPBP6 could be one?

It is indeed very intriguing that we never came across GTPBP6 in the present study as well as in previous biochemical studies (Cipullo et al., 2021 PMID: 33283228). This probably relies on the fact that GTPBP6 is indeed acting downstream of GTPBP5, as the reviewer mentioned. We have now discussed this aspect in the manuscript as follows (page 10, line 317-320):

“In particular, in Hillen et al. ³⁶, GTPBP6 is proposed to act at a later stage of assembly where MTERF4-NSUN4 and GTPBP5 are released, suggesting that GTPBP6 might be important for the release of these factors.”

Then together these two manuscripts read very well and the GTPBP6 manuscript follows neatly on from the GTPBP5 work. Finally, I liked the final assembly figure of Cipullo and was not massively wowed by the video in Heller. Could I ask that a similar figure be included to help the reader in Heller et al.?